# Diabetes screening: a pending issue in hypertense/obese patients

Armina Sepehri[1], Antonio Palazón-Bru[1], Vicente Francisco Gil-Guillén[1], Dolores Ramírez-Prado[1], Felipe Navarro-Cremades[1], Ernesto Cortés[2] and María Mercedes Rizo-Baeza[3]

[1] Department of Clinical Medicine, Miguel Hernández University, San Juan de Alicante, Alicante, Spain
[2] Department of Pharmacology, Paediatrics and Organic Chemistry, Miguel Hernández University, San Juan de Alicante, Alicante, Spain
[3] Department of Nursing, University of Alicante, San Vicente del Raspeig, Alicante, Spain

## ABSTRACT

The literature about possible cardiovascular consequences of diagnostic inertia in diabetes is scarce. We examined the influence of undetected high fasting blood glucose (FBG) levels on the cardiovascular risk and poor control of cardiovascular risk factors in hypertensive or obese patients, with no previous diagnosis of diabetes mellitus (i.e., diagnostic inertia). A cross-sectional study during a preventive program in a Spanish region was performed in 2003–2004. The participants were aged $\geq 40$ years and did not have diabetes but were hypertensive ($n = 5,347$) or obese ($n = 7,833$). The outcomes were high cardiovascular risk (SCORE $\geq 5\%$), poor control of the blood pressure ($\geq 140/90$ mmHg) and class II obesity. The relationship was examined between FBG and the main parameters, calculating the adjusted odd ratios with multivariate models. Higher values of FBG were associated with all the outcomes. A more proactive attitude towards the diagnosis of diabetes mellitus in the hypertensive and obese population should be adopted.

Subjects Diabetes and Endocrinology, Epidemiology, Nutrition, Public Health
Keywords Diabetes mellitus, Hypertension, Delayed diagnosis, Obesity, Physician's practice patterns, Primary health care

## INTRODUCTION

Cardiovascular diseases (CVD) are the leading cause of mortality in the world (*WHO, 2011*). Diabetes, along with hypertension, dyslipidaemia, obesity and smoking, is one of the main risk factors for CVD (*D'Agostino Sr et al., 2008*). Moreover, these risk factors are associated with each other; for instance, the chronic hyperglycaemia produced by diabetes is associated with high blood pressure (BP) and obesity (*Expert Committee on the Diagnosis and Classification of Diabetes Mellitus, 2003*).

Clinical inertia has been defined as failure by the health care provider to initiate or intensify the treatment when this is in fact indicated (*Phillips et al., 2001*). A very recent systematic review (*Lebeau et al., 2014*) determined that this concept has been adapted to making mistakes in the diagnosis of a disorder (rather than its treatment), and is known as diagnostic inertia. This diagnostic inertia has already been assessed in both hypertension and in dyslipidaemia, determining both its prevalence and its association

Corresponding author
Antonio Palazón-Bru,
antonio.pb23@gmail.com

with a personal history of cardiovascular risk factors (CVRF) in the patients studied (*Gil-Guillén et al., 2010*; *Palazón-Bru et al., 2014*). *Sepehri et al. (2014)* assessed the relation between inertia (defined as lack of advice about losing weight in obese patients) and the cardiovascular risk, measured with a probabilistic function. However, we have found no study examining the possible cardiovascular consequences of diagnostic inertia in diabetes. This concept is of great importance, as an early diagnosis is more likely to result in fewer future complications of CVD (*D'Agostino Sr et al., 2008*). Accordingly, we carried out a study analysing patients without a diagnosis of diabetes but with a diagnosis of hypertension or obesity to determine the existence or otherwise of an association between the lack of detection of high fasting basal plasma glucose (FBG) concentrations and a greater cardiovascular risk; i.e., assessing whether diagnostic inertia in diabetes mellitus is resulting in situations of high cardiovascular risk. Thus, the results provided here are novel and could show the need to evaluate measures to improve the diagnosis of diabetes in hypertensive and obese patients who attend the health centre. We hypothesised that diagnostic inertia is not associated with long-term cardiovascular consequences.

## MATERIALS & METHODS

### Setting

The Valencian Community is a Mediterranean region in eastern Spain with a population (January 2004) of 4,518,126 inhabitants. Primary health care is provided in health centres, has a universal coverage and has no cost for patients. The prevalence of diabetes is high and it is strongly associated with other known risk factors. Moreover, a large proportion of persons are not diagnosed, although recently screening for diabetes has greatly increased (*Generalitat Valenciana: Conselleria de Sanitat, 2010a*).

In this Community, at the end of 2003, a preventive activities program addressed to all persons aged 40 years or over was started. Each person was invited to participate by post, after which an appointment at their health centre was made by telephone. A preventive examination was made by physicians and nurses, and the person was given a report with the result of the examination and convenient recommendations, a copy of which remained at the health centre. This program followed the guidelines of the Spanish Society of Family and Community Medicine (*Villar Alvarez et al., 2003*; *Sepehri et al., 2014*). The program included screening for cardiovascular (hypertension, diabetes, dyslipidaemia, smoking, obesity...) and gynaecologic (cytology, mammogram,...) conditions and a vaccination campaign (flu, tetanus and pneumococcal).

### Study population

The study population comprised all persons who attended their health centre. The main characteristics of these persons were: predominantly women, coexistence of CVRF, older age and frequent visitors (*Pedrera Carbonell et al., 2005*).

### Study design and participants

This observational, cross-sectional study analyzed a sample of nondiabetic persons aged ≥40 years who had hypertension or were obese and who attended their health

centre during the first six months of the preventive activities program of the Valencian Community and who were willing to collaborate. A patient was considered to have diabetes/hypertension if their physician had previously diagnosed these conditions with codes 250/401 of ICD-9-CM. Obesity was defined as a body mass index (BMI) $\geq 30\,\text{kg/m}^2$. Any patient who was diabetic or who did not have either of the two disorders (hypertension or obesity) was excluded.

## Variables

The main parameters constructed were:

(a) Poor BP control ($\geq 140/90$ mmHg) in patients with hypertension (*Chobanian et al., 2003*).

(b) Class II obesity (BMI $\geq 35\,\text{kg/m}^2$) in obese patients (*WHO, 1997*).

(c) A SCORE $\geq 5\%$ (total cholesterol (TC) version) (*Conroy et al., 2003*). The SCORE is a probabilistic equation to estimate the 10-year risk of cardiovascular death in persons aged 40–65 years without prior CVD. The variables included in this model were: age, gender, TC, systolic BP (SBP), and smoking. This parameter was defined for both analyses (hypertension and obesity).

The following variables were also analyzed: gender, FBG group (<7.0 mmol/L, 5.6–7.0 mmol/L, <5.6 mmol/L) (*Expert Committee on the Diagnosis and Classification of Diabetes Mellitus, 2003*; *Genuth et al., 2003*), personal history of smoking or CVD (coronary heart disease (CHD) or stroke), SBP and diastolic BP (DBP) (mmHg), TC (mmol/L), age (years), and BMI ($\text{kg/m}^2$). Based on the personal history of CVD, the variable secondary cardiovascular prevention was defined as that patient who had already experienced a cardiovascular event (CHD or stroke) prior to starting the study. A patient was considered to be in primary cardiovascular prevention if they had no history of CVD at the time of entering the study.

## Measurements

The measurements of FBG and TC were made from first morning blood samples taken after a minimum 8-h fast and using calibrated devices. The personal history of diseases and smoking, the gender, and the age were ascertained at a patient interview and corroborated with the medical history. The BP was measured in accordance with current guidelines using well-calibrated semiautomated anaeroid—mercury—devices under suitable conditions. The BMI was calculated from the weight and height, measured with a calibrated scale and stadiometer after removing any objects that could affect the weight and with no shoes. The SCORE was computed using all the variables required for its calculation (*Conroy et al., 2003*).

## Sample size

The sample included 5,347 patients with hypertension (of whom 3,003 fulfilled the criteria for calculation of the SCORE) and 7,833 obese patients (SCORE in 5,827). We aimed to

determine whether there existed dependence (Pearson $X^2$ tests in a $3 \times 2$ table) between the FBG groups and the main parameters (control, class II obesity and high SCORE in patients with hypertension or obesity). As the sample was collected prior to calculating the sample size, we calculated the power of the contrast of the $X^2$ with the sample (*Chow, Wang & Shao, 2007*). Thus, we approximated the $X^2$ value with the data of 1,500 hypertensive and 1,500 obese patients selected randomly from the final sample. With these values and using a significance level of 5%, we obtained a Pearson $X^2$ contrast power >98% in all the analyses done.

## Statistical analysis

Absolute and relative frequencies are used to describe the qualitative variables, and means and standard deviations for the quantitative variables. Multivariate logistic regression models were used to estimate the adjusted odds ratio (ORs) in order to analyze the relation between the main parameters (poor control of the BP, class II obesity and high cardiovascular risk) and the other variables. For the analysis of the poor control of the BP and class II obesity, the ORs were adjusted for gender, FBG group, smoking history, SBP, DBP, TC, age, and BMI. In this analysis, we eliminated from the model those parameters that had collinearity with the control of the BP and class II obesity (SBP and DBP for hypertension, BMI for obesity). In the analysis of the high cardiovascular risk, the ORs were adjusted for FBG group and BMI. The other variables were not used, given their collinearity with the SCORE function (*Conroy et al., 2003*). The likelihood ratio test was used to determine the fit of the models. All the analyses were done with a significance level of 5% and the confidence intervals (CI) were calculated for the most relevant parameters. The analyses were all done using IBM SPSS Statistics 19.

## Ethical considerations

This study is based on an institutional agreement between the Conselleria de Sanitat and Miguel Hernández University, Elche (reference number: AVS-UV1.07X) authorized by an Institutional Review Board (Universidad Miguel Hernández de Elche-Conselleria de Sanitat de la Generalitat Valenciana). The data were therefore analyzed in compliance with current legislation on medical ethics. This institution played no role in data collection, analysis or interpretation, nor in the publication of the final manuscript. Additionally, the data were anonymized and encrypted, in accordance with the data protection law.

This population-based, non-interventional study (data from the Valencian Community) used data from medical records and informed consent was not required for included patients. The institutional agreement approved this consent procedure and ensured that information access was restricted, it did not compromise the interests or welfare of any patient, it minimized the risk of injury and its use was in line with current legislation.

## RESULTS

Of a total of 5,347 patients with hypertension, 58.2% (95% CI [56.9–59.5]) had their BP poorly controlled (Table 1). This poor control was significantly ($p < 0.05$) associated with: male gender (OR = 1.26, 95% CI [1.12–1.42]), FBG group ($\geq 7.0$ mmol/L, OR = 1.43, 95%

**Table 1** Analysis of fasting blood glucose screening results in hypertensive patients at primary healthcare centres in the Valencian Community (Spain), 2003–2004 data.

| Variable | Total 5,347 n(%)/x ± s | Poor BP control 3,113(58.2) n(%)/x ± sd | (Adj. OR) | 95% CI (Adj. OR) | Primary prevention ≤65 years 3,003 n(%)/x ± sd | High cardiovascular risk 278(9.3) n(%)/x ± sd | Adj. OR | 95% CI Adj. OR |
|---|---|---|---|---|---|---|---|---|
| **Gender** | | | | | | | | |
| Male | 2,111(39.5) | 1,288(61.0) | 1.26[d] | (1.12, 1.42) | 1,115(37.1) | 242(21.7) | N/A | N/A |
| Female[a] | 3,236(60.5) | 1,825(56.4) | | | 1,888(62.9) | 36(1.9) | | |
| **FBG groups** | | | | | | | | |
| ≥ 7.0 mmol/L | 399(7.5) | 264(66.2) | 1.43 | (1.15,1.79) | 193(6.4) | 23(11.9) | 1.56 | (0.97,2.51) |
| 5.6–6.99 mmol/L | 1,932(36.1) | 1,201(62.2) | 1.27 | (1.13,1.43) | 1,071(35.7) | 113(10.6) | 1.34 | (1.03,1.75) |
| < 5.6 mmol/L[a] | 3,016(56.4) | 1,648(54.6) | [d] | | 1,739(57.9) | 142(8.2) | [b] | |
| **Personal history of smoking** | | | | | | | | |
| Yes | 656(12.3) | 381(58.1) | 1.05 | (0.88,1.25) | 474(15.8) | 110(23.2) | N/A | N/A |
| No[a] | 4,691(87.7) | 2,732(58.2) | | | 2,529(84.2) | 168(6.6) | | |
| **Secondary prevention** | | | | | | | | |
| Yes | 673(12.6) | 357(53.0) | 0.73[d] | (0.62,0.86) | N/A | N/A | N/A | N/A |
| No[a] | 4,674(87.6) | 2,756(59.0) | | | | | | |
| SBP (mmHg) | 138.5 ± 16.4 | 148.4 ± 13.2 | N/A | N/A | 137.3 ± 16.3 | 152.2 ± 17.8 | N/A | N/A |
| DBP (mmHg) | 82.4 ± 9.7 | 86.7 ± 9.1 | N/A | N/A | 83.8 ± 9.8 | 88.4 ± 9.8 | N/A | N/A |
| TC (mmol/L) | 5.5 ± 0.9 | 5.5 ± 0.9 | 1.10[c] | (1.03,1.17) | 5.5 ± 0.9 | 5.7 ± 0.9 | N/A | N/A |
| Age (Years) | 61.8 ± 9.5 | 62.3 ± 9.3 | 1.02[d] | (1.01,1.02) | 55.9 ± 6.5 | 61.3 ± 3.8 | N/A | N/A |
| BMI (kg/m$^2$) | 29.9 ± 4.5 | 30.3 ± 4.6 | 1.05[d] | (1.03,1.06) | 30.0 ± 4.7 | 30.0 ± 4.1 | 0.99 | (0.97,1.02) |

**Notes.**

n(%), absolute frequency (relative frequency); x ± sd, mean ± standard deviation; BP, blood pressure; Adj. OR, adjusted odds ratio; CI, confidence interval; FBG, fasting blood glucose; SBP, systolic blood pressure; DBP, diastolic blood pressure; TC, total cholesterol; BMI, body mass index; N/A, not applicable.

[a] Reference.

[b] $0.01 < p < 0.05$.

[c] $0.001 < p < 0.01$.

[d] $p < 0.001$.

Likelihood ratio test for poor control of the BP: $X^2 = 135.2$ $p < 0.001$; primary prevention ≤65 years: $X^2 = 72.9$ $p < 0.001$.

In the poor control analysis, OR were adjusted for gender, FBG groups, smoking, secondary prevention, TC, age, and BMI. SBP and DBP were not included in the multivariate model due to collinearity with the control of the BP.

In primary prevention ≤ 65 years, OR were adjusted for FBG groups and BMI. Gender, smoking, SBP, TC and age were not included in the multivariate model due to collinearity with the cardiovascular risk. DBP was not included because of the collinearity with the SBP. Secondary prevention was not included in the model because the analysis was only in primary prevention patients.

CI [1.15-1.79]; 5.6–6.99 mmol/L, OR = 1.27, 95% CI [1.13–1.43]; <5.6 mmol/L, OR = 1), not being in secondary prevention (OR = 0.73, 95% CI [0.62–0.86]), high TC (OR = 1.10, 95% CI [1.03–1.17]), older age (OR = 1.02, 95% CI [1.01–1.02]), and BMI (OR = 1.05, 95% CI [1.03–1.06]). Figure 1A shows more clearly the relation between FBG level and poor BP control; higher levels of FBG were associated with a greater likelihood of poor control of the BP.

Table 1 also shows the analysis in the 3,003 patients who met the criteria for calculation of the SCORE. The magnitude of a high cardiovascular risk was 9.3% (95%

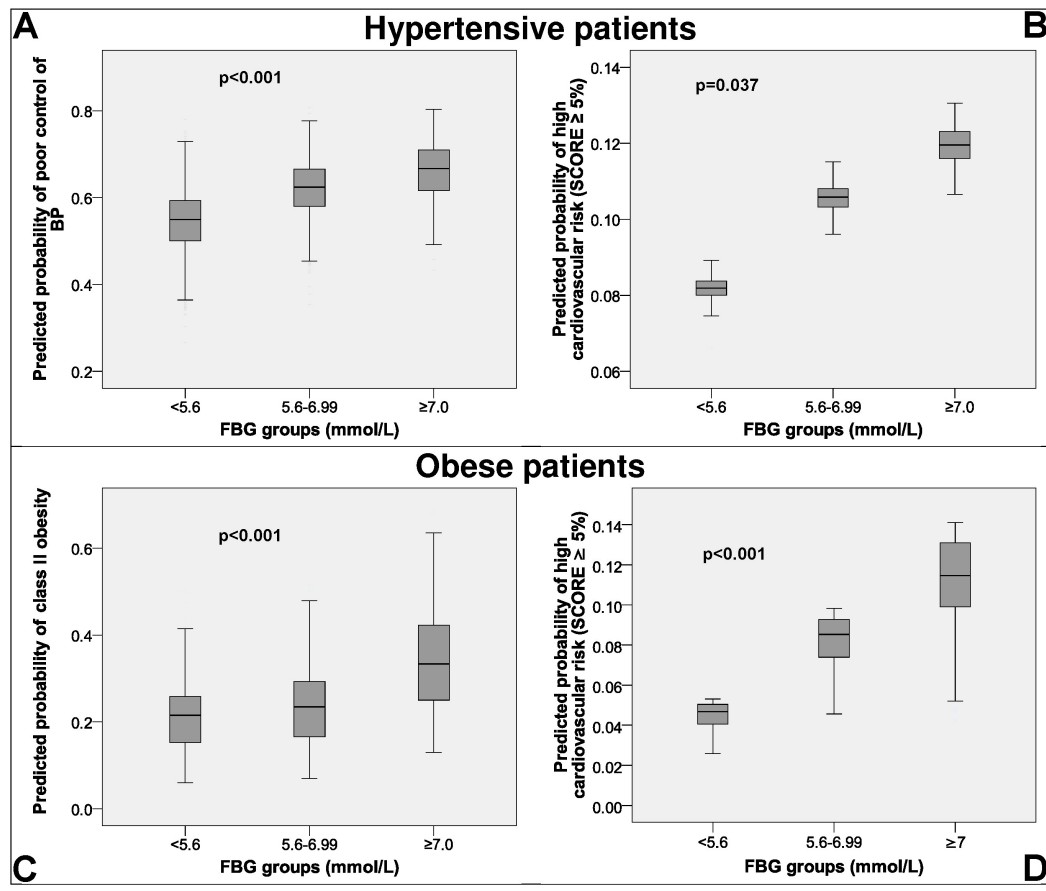

**Figure 1** **Predicted probability of high cardiovascular risk, poor control of hypertension and class II obesity. 2003–2004 data.** FBG, fasting blood glucose; BP, blood pressure. (A) Predicted probability of poor blood pressure control in hypertense patients. (B) Predicted probability of high cardiovascular risk in hypertense patients. (C) Predicted probability of class II obesity in obese patients. (D) Predicted probability of high cardiovascular risk in obese patients.

CI [8.2–10.3%]), and it was significantly associated with the FBG group ($\geq$7.0 mmol/L, OR = 1.56, 95% CI [0.97–2.51]; 5.6–6.99 mmol/L, OR = 1.34, 95% CI [1.03–1.75]; <5.6 mmol/L, OR = 1). Figure 1B shows an increasing relation between FBG level and a greater probability of having a high cardiovascular risk.

Of a total of 7,833 obese patients, 1,783 had a BMI $\geq$ 35 kg/m$^2$ (22.8%, 95% CI [21.8–23.7]) (Table 2). The factors significantly associated with class II obesity were: female gender (OR = 0.46, 95% CI [0.41–0.52]), FBG group ($\geq$7.0 mmol/L, OR = 2.11, 95% CI [1.73–2.57]; 5.6–6.99 mmol/L, OR = 1.22, 95% CI [1.08–1.36]; <5.6 mmol/L, OR = 1), high SBP (OR = 1.01, 95% CI [1.01–1.02]), and younger age (OR = 0.98, 95% CI [0.97–0.99]). As with hypertension, Fig. 1C also shows an increasing relation between FBG levels and class II obesity.

The analysis of the 5,827 obese patients who had the criteria necessary to calculate the SCORE showed a high SCORE in 6.2% (95% CI [5.6–6.8%]). The variables significantly associated with this problem were the FBG levels ($\geq$7.0 mmol/L, OR = 2.93, 95% CI

**Table 2  Analysis of fasting blood glucose screening results in obese patients at primary healthcare centres in the Valencian Community (Spain), 2003–2004 data.**

| Variable | Total 7,833 $n(\%)/x \pm sd$ | Class II obesity 1,783(22.8) $n(\%)/x \pm sd$ | Adj.OR | 95% CI (Adj. OR) | Primary prevention ≤ 65 years 5,827 $n(\%)/x \pm sd$ | High cardiovascular risk 360(6.2) $n(\%)/x \pm sd$ | Adj. OR | 95% CI (Adj. OR) |
|---|---|---|---|---|---|---|---|---|
| **Gender** | | | | | | | | |
| Male | 3,190(40.7) | 513(16.1) | 0.46[d] | (0.41, 0.52) | 2,365(40.6) | 242(21.7) | NA | NA |
| Female[a] | 4,643(59.3) | 1,270(27.4) | | | 3,462(59.4) | 36(1.9) | | |
| **FBG groups** | | | | | | | | |
| ≥ 7.0 mmol/L | 550(7.0) | 187(34.0) | 2.11 | (1.73,2.57) | 357(6.1) | 23(11.9) | 2.93 | (2.02,4.25) |
| 5.6–6.99 mmol/L | 2,808(35.8) | 659(23.5) | 1.22 | (1.08,1.36) | 2,058(35.3) | 113(10.6) | 1.95 | (1.55,2.44) |
| <5.6 mmol/L[a] | 4,475(57.1) | 937(20.9) | [d] | | 3,412(58.6) | 142(8.2) | [d] | |
| **Personal history of smoking** | | | | | | | | |
| Yes | 1,380(17.6) | 273(19.8) | 0.92 | (0.78,1.07) | 1,222(21.0) | 110(23.2) | N/A | N/A |
| No[a] | 6,453(82.4) | 1,510(23.4) | | | 4,605(79.0) | 168(6.6) | | |
| **Secondary prevention** | | | | | | | | |
| Yes | 503(6.4) | 118(23.5) | 1.23[b] | (0.98,1.53) | N/A | N/A | N/A | N/A |
| No[a] | 7,330(93.6) | 1,665(22.7) | | | | | | |
| SBP (mmHg) | 133.9 ± 17.0 | 136.0 ± 17.9 | 1.01[d] | (1.01,1.02) | 132.2 ± 16.8 | 150.3 ± 19.2 | N/A | N/A |
| DBP (mmHg) | 81.4 ± 10.1 | 82.9 ± 10.6 | N/A | N/A | 81.6 ± 10.2 | 87.5 ± 10.5 | N/A | N/A |
| TC (mmol/L) | 5.5 ± 1.0 | 5.5 ± 0.9 | 0.95[b] | (0.90,1.01) | 5.5 ± 1.0 | 5.8 ± 1.0 | N/A | N/A |
| Age (Years) | 56.7 ± 9.9 | 55.9 ± 9.7 | 0.98[d] | (0.97,0.99) | 52.6 ± 7.3 | 60.6 ± 4.3 | N/A | N/A |
| BMI (kg/m$^2$) | 33.3 ± 3.2 | 38.0 ± 3.0 | N/A | N/A | 33.4 ± 3.3 | 32.9 ± 2.7 | 0.94[c] | (0.98,0.98) |

**Notes.**

$n$ (%), absolute frequency (relative frequency); $x \pm sd$, mean ± standard deviation; Adj. OR, adjusted odds ratio; CI, confidence interval; FBG, fasting blood glucose; SBP, systolic blood pressure; DBP, diastolic blood pressure; TC, total cholesterol; BMI, body mass index; N/A, not applicable.

[a] Reference.

[b] $0.05 < p < 0.1$.

[c] $0.001 < p < 0.01$.

[d] $p < 0.001$.

Likelihood ratio test for class II obesity: $X^2 = 290.0 \ p < 0.001$; primary prevention ≤65 years: $X^2 = 56.8 \ p < 0.001$.

In the poor control analysis, OR were adjusted for gender, FBG groups, smoking, secondary prevention, SBP, TC, and age. BMI was not included in the multivariate model due to collinearity with the control of the obesity. DBP was not included because of the collinearity with the SBP.

In primary prevention ≤65 years, OR were adjusted for FBG groups and BMI. Gender, smoking, SBP, TC and age were not included in the multivariate model due to collinearity with the cardiovascular risk. DBP was not included because of the collinearity with the SBP. Secondary prevention was not included in the model because the analysis was only in primary prevention patients.

[2.02–4.25]; 5.6–6.99 mmol/L, OR = 1.95, 95% CI [1.55–2.44]; <5.6 mmol/L, OR = 1). Of note was the association between the increase in FBG level and a greater cardiovascular risk (Fig. 1D).

## DISCUSSION

The analysis of the patients with hypertension or obesity showed a direct relation between the levels of FBG, and poor control of the BP and class II obesity, as well as with a greater risk for death due to CVD. A search of the relevant literature produced just one study evaluating the association in obese patients between cardiovascular risk (measured with

a calibration of the Wilson scale of the Framingham Study) and inertia, considered to be the absence of individualized advice to lose weight (*Sepehri et al., 2014*). The authors of this study found the same association as us. Although other studies have related inertia with poor control of CVRF, this inertia concerned persons who had already been diagnosed with these CVRF (*Okonofua et al., 2006*). This indicates that there exists a lack of studies addressing this topic.

When we started this study we expected to find that our patients who had no previous diagnosis of diabetes would have a lower magnitude of altered FBG screening results and that this would not be associated with a greater cardiovascular risk, poor BP control or class II obesity. However, the results for this association were not expected, as all these patients attended their health centre regularly. At these visits to control their CVRF, the physicians ought to order a blood test to control the lipid and FBG levels in order to discard the presence of dyslipidaemia and diabetes mellitus, respectively, as these CVRF usually coexist in patients with hypertension and obesity (*Pedrera Carbonell et al., 2005*).

A possible explanation for detecting a greater cardiovascular risk in these hypertensive or obese patients with no previous diagnosis of diabetes but with high FBG levels may relate to the underestimation of the true cardiovascular risk, as healthcare professionals are reported to use the risk functions recommended in the various clinical guidelines inadequately (*Banegas et al., 2006*; *Márquez-Contreras et al., 2007*).

Concerning the associations found between gender and the various outcomes, the men had worse control of their hypertension, possibly because men generally attend their health care centre less frequently for control of their CVRF, which could result in worse BP control (*Pedrera Carbonell et al., 2005*). The women, however, were more likely to have class II obesity. This, though, was expected and logical, as the prevalence of class II and III obesity is higher in women than men in our region (*Generalitat Valenciana: Conselleria de Sanitat, 2010b*).

Healthcare policy should participate actively in the fight against diabetes mellitus via its early detection and control, incorporating alarms in the electronic records systems when a FBG measurement gives abnormal results or when a patient with one or more CVRF has failed to have a FBG measurement for a certain number of months. Our results show the need to integrate these healthcare policies in health centres to enable the early detection of diabetes mellitus in the hypertensive or the obese patient. This will then encourage physicians and nurses to attempt to control the disease thereby reducing the incidence of CVD, as the most beneficial aspects of treatment of diabetes mellitus are seen during the early stage of the disease (*Schernthaner, 2010*).

## Strengths and limitations of the study

The main strength of this study is that it is the first to look at the possible cardiovascular consequences of committing diagnostic inertia in patients with diabetes mellitus who have a diagnosis of hypertension or obesity. The results therefore are novel. In addition, the statistical power of the study is enhanced by the large sample size. This minimizes the random error when drawing conclusions about a population attending their health centre.

Moreover, the fact that all the health centres in the Valencian Community participated in the study provides external validity. Furthermore, the contrast power used was over 98%, when studies generally calculate their sample sizes from tables indicating an 80% or 90% power.

The limitations are related to those found in any cross-sectional study measuring the magnitude of a problem; that is, they are unable to produce any longitudinal or causal findings. Concerning bias, the most important is selection bias, given the characteristics of the study sample. This bias concerns the fact that it is the most motivated patients who request a preventive check-up. For obvious reasons this cannot be changed, as each person has a different degree of motivation. However, this does not affect the purpose of the study, which was to assess the lack of detection of high FBG concentrations in patients with no diagnosis of diabetes but who did have a diagnosis of hypertension or obesity. As for measurement bias, all the teams were requested to use reliable devices to measure the variables and to do the clinical interview correctly. Logically, this bias too is accepted in this type of study. Finally, although we used 2003–2004 data, we have to take into account that diagnostic inertia is a prevalent problem which was already detected in the 1980s (*Bell & Dippe, 1988*); however, it was not named inertia, and even nowadays it is being studied (*Palazón-Bru et al., 2014*). During these last decades the clinical guidelines have been updated and certain cut-off control parameters have been modified, but diagnostic inertia is still taking place.

## CONCLUSIONS

This study provides useful information for clinical practice about the influence of the lack of detection of high FBG concentrations in obese or hypertensive patients with no previous diagnosis of diabetes mellitus; i.e., the possible repercussions of committing diagnostic inertia in patients with diabetes. The FBG has a direct negative influence, associated with a greater cardiovascular risk and worse control of BP and class II obesity. Nonetheless, the results should be interpreted with caution until such time as other studies are able to corroborate our findings.

The main learning point arising from this study is that we have to adopt a more proactive attitude towards the diagnosis of diabetes mellitus in patients who are obese or who have hypertension. This attitude should centre on the early detection of diabetes, greater use of cardiovascular risk scales, and the incorporation of alert systems in the electronic medical records of patients attending their healthcare professional to minimize the problem and control the FBG in nondiabetic persons to detect the disease as early as possible.

## ACKNOWLEDGEMENTS

The authors thank Ian Johnstone for help with the English language version of the text.

### Funding

The Conselleria de Sanitat (Valencian Community) gave permission and financial support for this study. This public organism subsidized and authorized this study exclusively to determine the situation in patients attending their health center. It played no part in the study design, data collection, analysis or interpretation, writing the manuscript or the decision to send it for publication. The grant was used to contract a person (Antonio Fernández) to collect and computerize the data. The funders had no role in study design, data collection and analysis, decision to publish, or preparation of the manuscript.

### Grant Disclosures

The following grant information was disclosed by the authors:
Conselleria de Sanitat (Valencian Community).

### Competing Interests

The authors declare there are no competing interests.

### Author Contributions

- Armina Sepehri and Vicente Francisco Gil-Guillén conceived and designed the experiments, analyzed the data, wrote the paper, reviewed drafts of the paper.
- Antonio Palazón-Bru conceived and designed the experiments, analyzed the data, wrote the paper, prepared figures and/or tables, reviewed drafts of the paper.
- Dolores Ramírez-Prado, Felipe Navarro-Cremades, Ernesto Cortés and María Mercedes Rizo-Baeza conceived and designed the experiments, reviewed drafts of the paper.

### Human Ethics

The following information was supplied relating to ethical approvals (i.e.,, approving body and any reference numbers):

This study is based on an institutional agreement between the Conselleria de Sanitat and Miguel Hernández University, Elche (reference number: AVS-UV1.07X) authorized by an Institutional Review Board (Universidad Miguel Hernández de Elche-Conselleria de Sanitat de la Generalitat Valenciana). The data were therefore analyzed in compliance with current legislation on medical ethics. This institution played no role in data collection, analysis or interpretation, nor in the publication of the final manuscript. Additionally, the data were anonymized and encrypted, in accordance with the data protection law.

This population-based, non-interventional study (data from the Valencian Community) used data from medical records and informed consent was not required for included patients. The institutional agreement approved this consent procedure and ensured that information access was restricted, it did not compromise the interests or welfare of any patient, it minimized the risk of injury and its use was in line with current legislation.

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
