# Peer review of "Diabetes screening: a pending issue in hypertense/obese patients"

_PeerJ, doi:10.7717/peerj.914_

## Round 0.1 · original submission · Major Revisions

Dear Dr. Rizo-Baeza,

The paper has been reviewed by two experts in the field and major concerns have been raised about your study.

Reviewer 1 ·

Basic reporting

The submission do not adhere to PeerJ policies, e.g. Ethical Statement, because the Agreement between 2 Institutions the Conselleria de Sanitat and Miguel Hernandez University is insufficient. Ethical committee should approve the study. The article English is not clear and do not conform to professional standards of courtesy and expression. The article includes insufficient introduction and background. On the other hand there is redundant information about the diagnostic threshold for diabetes. Another sentence at the end of first page of introduction: “The Valencian Community is a Mediterranean region in eastern Spain…..” This sentence belongs to the Method section. Relevant prior literature is missing, e.g. Lang et al 2015 reported clinical inertia in glycemic control among with Type 2 diabetes. The structure of the submitted article conforms to the templates. Figures are relevant to the content of the article, of sufficient resolution, and appropriately described and labeled. Relevant hypothesis is missing. The bodies of work is not coherent.

Experimental design

The submission don´t describe original primary research within the Scope of the journal. Authors aimed to assess the clinical inertia in diabetes effect on increased cardiovascular risk. However they did not evaluated this clinical inertia. Also the description of results in abstract is confusing. Submission has not clearly defined the research question. The investigation has not been conducted rigorously and do not have a high technical standard. Methods are not described with sufficient information. The numbers of patients in the 2 groups with hypertension and group of patients with obesity vary according the fact if they had SCORE or not calculated. Authors should elucidate this discrepancy. What does it mean “poor control of obesity”? Do we have good and poor control of obesity??? The devices for blood pressure, glucose, TC,… measurement used in this study were not described sufficiently. Secondary prevention visits were not mentioned in the method section, as well as the time between first and second visit. The text is sometimes lacking explanation of used abbreviations (e.g. method section TC).

Validity of the findings

The data are not robust, statistically sound, and controlled. The conclusions are not appropriately stated, they are not connected to the original question investigated. There is a lack of novelty.

Additional comments

The scientific aim was set up interestingly, however finally authors did not evaluated clinical inertia of diabetes and their effects on cardiovascular risk.

·

Basic reporting

No comments.

Experimental design

No comments.

Validity of the findings

No comments.

Additional comments

Cardiovascular diseases are a major cause of death worldwide, and diabetes, hypertension, dislipidemia and obesity constitute some of its main risk factors (either alone and/or as components of the so-called metabolic syndrome). Given the preventive activities program established by the Valencian Community (Spain) to people over 40 years old and the lack of evidences on the possible cardiovascular consequences of clinical inertia in the diabetes diagnosis, these authors analysed herein if diagnostic inertia in diabetes (despite a diagnosis of hypertension or obesity) could result in a higher cardiovascular risk. This study emphasizes that a more proactive attitude should be taken towards the diagnosis of diabetes mellitus in patients who are obese or hypertensive.
Although this is a highly valuable contribution to the early and successful management of diabetes and its related cardiovascular risk (particularly in obese or hypertensive patients), I have a few minor issues that should be addressed before it is acceptable to be published:

General comments:
- Please, normalize the references throughout the main text.

Discussion:
- Do the authors have any explanation for the male gender-associated poorer blood pressure control and for the lack of association with aging described in Table 1? A similar question can be pose on the association between female gender and obesity (Table 2).

---

## Round 0.2 · accepted · Accept

Dear Dr. Rizo-Baeza,

I am pleased to inform you that your manuscript has been accepted for publication in PeerJ.

Reviewer 1 ·

Basic reporting

The submission adhere to all PeerJ policies.

Experimental design

The article meet the standards of PerJ

Validity of the findings

The data are robust, statistically sound, and controlled.

Additional comments

Manuscript is suitable for publication now.

·

Basic reporting

No further comments.

Experimental design

No further comments.

Validity of the findings

No further comments.